# Development of a Positive Psychology Well-Being Intervention in a Community Pharmacy Setting

**DOI:** 10.3390/pharmacy11010014

**Published:** 2023-01-11

**Authors:** Jennifer Louise Ward, Alison Sparkes, Marie Ricketts, Paul Hewlett, Amie-Louise Prior, Britt Hallingberg, Delyth Higman James

**Affiliations:** 1Department of Applied Psychology, Cardiff School of Sport and Health Sciences, Cardiff Metropolitan University, Cardiff CF5 2YB, Wales, UK; 2The Health Dispensary, 135-155 Windsor Rd., Neath SA11 1N, Wales, UK

**Keywords:** community pharmacy, well-being, positive psychology, public health, intervention, PERMA, diary

## Abstract

**Background**: Community pharmacies are well-placed to deliver well-being interventions; however, to date, nothing has been produced specifically for this setting. The aim of this study was to develop a positive psychology intervention suitable for a community pharmacy setting with the goal of increasing the well-being of community members. **Methods**: Intervention development consisted of three steps: Step 1—identify the evidence-base and well-being model to underpin the basis of the intervention (Version 1); Step 2—model the intervention and gather user feedback to produce Version 2, and Step 3—revisit the evidence-base and refine the intervention to produce Version 3. **Results**: Findings from nine studies (seven RCTs, one cross-sectional, one N-1 design plus user feedback were applied to model a 6-week ‘*Prescribing Happiness* (*P*-*Hap*)’ intervention, underpinned by the PERMA model plus four other components from the positive psychology literature (*Three Good Things*, *Utilising Your Signature Strengths in New Ways*, *Best Possible Selves and Character Strengths*). A PERMA-based diary was designed to be completed 3 days a week as part of the intervention. **Conclusions**: This work is an important development which will direct the future implementation of interventions to support well-being in this novel setting. The next stage is to gain the perspectives of external stakeholders on the feasibility of delivering the P-Hap for its adoption into community pharmacy services in the future.

## 1. Introduction

### 1.1. Community Pharmacy

Community pharmacies are important healthcare settings, strategically located to deliver services aimed at promoting health and preventing disease [1]. Community pharmacists play a key role in delivering services aimed at improving the health and well-being of their local population [2] with good customer relationships seen as one of their main strengths [3]. Public health services are inclusive of interventions that enhance both mental and physical health. In the United Kingdom (UK), many public health services have been integrated into health professional practices including General Practice (GP) surgeries and community pharmacies. The integration of public health services into community pharmacies diversifies what they can offer to their local communities. For example, community pharmacies can offer advice on the modification of health-related behaviours to minimise the risk of disease, screening for disease prevention and involvement in health promotion campaigns [4,5,6]. The National Institute for Health and Care Excellence (NICE) [7] advocates that community pharmacies are well placed to be developed into health and well-being hubs by offering health and well-being advice, delivering brief interventions, integrated into existing care and referral pathways.

### 1.2. Promoting Health and Well-Being in Community Pharmacy

In 2009, The ‘Healthy Living Pharmacy’ (HLP) programme was implemented in community pharmacies in England as a commissioned NHS Enhanced Service [7,8,9,10]. ‘Healthy Living Champions’ (HLCs) provide health information and signposts patients to relevant health services to enable them to lead healthier lifestyles. The promotion of behaviour change for better health and well-being are important aspects of the HCL’s training. This is the closest example of a community pharmacy-based well-being intervention in the UK; however, this programme is not specific to promoting psychological well-being and limited to England. However, other initiatives exist which promote the role of community pharmacy in supporting people with mental health conditions.

### 1.3. Support for Mental Health in Community Pharmacy

Mental Health First Aid (MHFA) training was first developed in Australia with the aim of reducing stigma and increasing knowledge, skills, and confidence in responding to acute mental health disorders [11]. The training is now available in over twenty-six countries including all four devolved nations of the UK [12]. Kirschbaum and colleagues explored the MHFA training needs of rural pharmacists in Australia and the barriers for providing effective mental health support [13]. They reported that developing and maintaining good relationships with patients was one of the most important factors for its successful implementation. Shams and Hattingh [14] evaluated the impact of the MHFA training across four towns in Australia. Community pharmacists reported having a better understanding of mental health issues, recognising the signs and symptoms earlier and felt more confident engaging in mental health conversations [14]. This highlights the potential for community pharmacies to play an important role in providing mental health support.

Rubio-Valera et al. reviewed the global evidence for pharmacist-delivered services in mental health care and found that pharmacists in primary and secondary care possess the necessary skills, knowledge, and attitudes for this role [15]. The authors also concluded that community pharmacists in the UK have well-established relationships with their patients and good communication channels with other healthcare providers to provide mental health support.

Whilst evidence exists to support the role of community pharmacists in dealing with mental health disorders, to date, there is no published literature that describes the development of community pharmacy-based interventions to support the holistic wellbeing (psychological, emotional, and social) of individuals who do not have an existing mental health diagnosis.

### 1.4. Well-Being and Positive Psychology

Positive psychology is a discipline introduced by Seligman in the United States of America (USA) in the late 1990s which saw a shift from conceptualising mental health only as a disease with a given pathology to a more positive approach. A state of positive well-being is the goal of positive psychology, which encompasses feeling good, having a sense of meaning, building good relationships, and functioning effectively [16]. A wide range of other aspects are included, such as hope, character strengths, flow, happiness, savouring, and resilience [17]. It also highlights factors that build strengths and help people to flourish which underlies optimal human functioning [18,19,20]. Positive psychology has been translated into effective evidence-based interventions that promote positive well-being, called positive psychology interventions (PPIs). PPIs can focus on a broad range of activities such as writing gratitude diaries [21], delivering gratitude letters [22] and practicing visualising ones’ best self [23,24].

### 1.5. Types of Well-Being

Two important aspects of well-being are hedonic and eudaiomonic well-being. Hedonic well-being is associated with pleasure and happiness [25,26]. The terms happiness and subjective well-being are often used inter-changeably; however, happiness is only related to the hedonic aspect of well-being [27,28]. The eudaiomonic view of well-being focuses on the experience of meaning and the realisation of one’s true potential [29,30]. Eudaiomonic well-being is driven by challenges and related to activities which encourage personal development, growth, and long-term well-being [31]. It has been suggested that the pursuit of happiness (hedonic) alone is not a worthwhile goal since psychological well-being (eudaiomonic) is achieved through pursuing meaningful and valuable lives [32]. The balanced pursuit of hedonic and eudaiomonic well-being is, therefore, essential to achieving and sustaining well-being.

### 1.6. Positive Psychology and Well-Being Interventions in Community Pharmacy

Policy makers have suggested that community pharmacies could play an important role in supporting good mental health and providing care to improve well-being [7,33,34]. To date, no well-being interventions have been developed to support public well-being specifically for delivery in a community pharmacy or empirically tested in this setting. This paper describes the processes of developing a positive psychology (well-being) intervention suitable for delivery in a community pharmacy setting to increase well-being in a sample of community members.

## 2. Materials and Methods

### 2.1. Study Design

The Medical Research Council (MRC) Framework for complex interventions [35] has a set of guidelines to support developing and evaluating interventions. The four phases include (i) Development, (ii) Feasibility, (iii) Evaluation, and (iv) Implementation. This study focused on Phase 1 ‘Development’ to support the production of a positive psychology (well-being) intervention which was appropriate for delivery in a community pharmacy setting.

Three specific research objectives were:To examine the evidence base from the literature and identify an appropriate underpinning theory (or model) to use as the basis for the development of Version 1 (V1) of a well-being intervention (Step 1),To model V1 and gather the perspectives of individuals on its design, content, and structure and to refine the intervention to produce Version 2 (V2) (Step 2), andTo revisit the evidence base by updating the literature search and further refine the intervention to produce Version 3 (V3) (Step 3).

An iterative process was adopted to address these objectives in accordance with the MRC framework guidelines [35] which recommend gathering the evidence and modelling the process before piloting or implementing an intervention in practice. This paper, therefore, reports this initial development phase only which was conducted between January 2018 and March 2019.

### 2.2. Setting

This research was centred on one independent community pharmacy located in the centre of a small town in a rural part of South Wales, UK. The population is approximately 143,300 [36] living in an area of social deprivation due to low incomes and lack of work, contributing to poor health and well-being [37]. The statistics for several health indicators for this geographic location show a high degree of health poverty, where deaths from suicide, premature deaths from heart disease and the consumption of alcohol are higher than the rest of Wales. It is, therefore, important that areas such as this have access to a wide range of healthcare services to support community health and well-being.

The pharmacy serves approximately 12,500 service users. The promotion of health and well-being is central to the ethos of the pharmacy which provides several free NHS Enhanced Services (e.g., supervised consumption of methadone, smoking cessation, emergency hormonal contraception, common ailment service and flu vaccination) in addition to other (Essential and Advanced) services. The volume of NHS prescriptions dispensed is estimated to be 12,500 per month with antidepressants featuring in the ten most frequently dispensed medications. Weight management, hearing checks and other health screening checks are available to purchase. A unique feature of pharmacy is that it also provides a counselling service and complementary healthcare services such as osteopathy, acupuncture, reflexology, reiki massage and aromatherapy massage. In total, around 500 of their clients utilise and pay for these services. The pharmacy has a total of twelve employees including two Pharmacists, one Accredited Checking Technician (ACT), three Pharmacy Technicians, three Dispensers, two Support Dispensers and one Delivery Driver. This pharmacy was therefore considered to be a suitable setting for the development of this novel well-being intervention since other non-NHS-funded services were already integrated into the business model.

### 2.3. Development of the Intervention


*Step 1–Examining the Evidence-Base/Identifying a Model to Produce Version 1*


PsychInfo, PubMed, and Scopus databases were searched from January 1999 to March 2018 to examine the literature and identify an appropriate model to form the basis of the well-being intervention. Three main areas of the literature were searched: positive psychology, public health services and community pharmacy interventions. The PICO (Participant, Intervention, Comparison, Outcomes) framework [38] was used to guide the literature search strategy. The inclusion and exclusion criteria, search strategy, keywords, approach, and interpretation of statistical analyses are presented in Appendix A.

The PRISMA statement for reporting systematic reviews that evaluate healthcare interventions was adopted [39]. The effectiveness of PPIs on well-being was examined in terms of short-term effects (post-intervention, 1 week, 2 weeks, and 1 month) and sustainable effects (3- and 6 months).

The methodological quality of studies was assessed using the 6-item scale established by the Cochrane collaboration tool [38]. The criteria were: (1) Adequacy of randomization concealment, (2) Blinding of subjects to the condition (blinding of assessors was not applicable in most cases), (3) Baseline comparability, (4) Power analysis: is there an adequate power analysis and/or are there at least 50 participants in the analysis? (5) Completeness of follow-up data: clear attrition analysis and loss to follow-up <50%, (6) Handling of missing data: the use of intention-to-treat analysis. The following scoring system was used; high when five to six criteria were met, medium when three to four were met, and low when zero, one or two were met.

The standardised mean effect sizes were calculated utilising the Wilson Practical Meta-Analysis Effect Size Calculator to understand the magnitude of intervention effectiveness across different timepoints (i.e., post-intervention, 1-week, 2-weeks, 1-, 3- and/or 6 months) on well-being. Effect sizes were assessed as small (Cohen’s d = 0.2), medium (Cohen’s d = 0.5) and large (Cohen’s d= 0.8) [40,41].

The researcher (J.L.W) also undertook observational activities at the pharmacy during the development process to consider any contextual issues relating to this specific environment, such as infrastructure, footfall, workflow, staff roles, and busy periods of activity. Notes kept by the researcher were incorporated into the development of V1. The perspectives of staff working in the research community pharmacy setting were also incorporated at this stage. In addition, a range of other types of community pharmacies were visited by the researcher to gain an understanding of the general characteristics of this setting. These factors were considered in the development of V1 of the intervention.


*Step 2—Modelling Version 1/Gathering Feedback to Produce Version 2*


V1 of the intervention was trialled with individuals including members of the research team (D.H.J, A-L.P, P.H, A.S), pharmacy staff and other individuals known to the researcher representing members of the public. The intervention was delivered on a one-to-one basis or as a small group once a week for a period of 7-weeks. V1 of the intervention was delivered (by J.L.W) either face-to-face or via a WhatsApp video call at a convenient time. Individuals were also asked to complete a diary every day of the week for a period of 7-weeks. Individual and group feedback was gathered (by J.L.W) at the end of each session to gain their perspectives on the content, delivery, diary, and overall intervention design. Based on this, decisions about any necessary were made by key members of the research team (J.L.W, D.H.J, A.S) to produce V2.


*Step 3—Revisiting the Evidence-base/Refining Version 2 to Produce Version 3*


The literature search undertaken in Step 1 was revisited to identify any newly published work. PsychInfo, PubMed, and Scopus databases were searched from April 2018 to March 2019 replicating the search strategy employed in Step 1. Based on this, members of the research team (J.L.W, D.H.J, A.S) made decisions about any necessary changes to produce V3.

## 3. Results

A total of three versions of the intervention were produced following the iterative process adopted in Steps 1 to 3.


*Step 1—Examining the Evidence Base/Identifying a Model for Version 1*


A total of nine studies were identified which investigated the short-term effects of PPIs delivered in community-based samples [42,43,44,45,46,47,48,49,50]. No studies were undertaken in a community pharmacy setting. The PRISMA four-phase diagram completed is illustrated in Appendix A.

The Positive emotions, Engagement, Relationships, Meaning and Accomplishments (PERMA) model of well-being [51] was identified as an appropriate underpinning model upon which to base the intervention (Table 1).

Each PERMA element [51] was mapped onto one or more of the intervention sessions. Reasons in support of using the PERMA model are as follows:It is a multi-component model which incorporates evidence-based constructs from positive psychology (gratitude, hope, flow, optimism, passion, and meaning) [43].It includes both hedonic and eudemonic well-being whilst other positive psychology models focus solely on one type of well-being (e.g., Authentic Happiness).It has been applied in a wide range of settings including community settings [42,43,44,45,46,47,48,49,50].It has been advocated as a stepping-stone to building well-being as it does not have an exhaustive list of elements [51].It is a simple model with straightforward terminology; therefore, its application was deemed feasible to be implemented with members of the public in a socially deprived area.

Description of Studies: (See Appendix A for a summary and detailed description of studies). With regard to research study design, seven out of the nine studies were randomised controlled trials (RCTs) [42,43,45,46,47,48,49] while one adopted a cross-sectional longitudinal study design [44] and one used a N-1 counterbalanced design [50]. Four studies utilised the *Early Memories* as a control group [42,47,48,49], four studies used control groups with no activity assigned [44,45,46,50] and one study used a placebo control exercise where the written activity was focused on describing different places the participants had passed that day [43]. All study participants were predominately female. The majority of interventions were delivered as 1-week online PPIs, which included one or two of the following PPIs: *Three Good Things, Utilising your Signature Strengths in New Ways, Gratitude Visit* and/or *Best Possible Selves*. A variety of different timeframes were used to measure the effect of PPIs on well-being. Four studies measured immediate post-intervention effects on well-being [44,45,46,50] whilst five studies measured the sustainability of the effect of the PPI on well-being at 3 months [42,43,47,48,49]. Further four studies measured the effect of the PPI on well-being for up to 6 months [42,47,48,49]. The attrition rates post-intervention varied from 0% [44,50] to 60% [43].

Effectiveness of PPIs on Well-being: (Appendix A): At post-intervention, all studies reported a statistically significant effect on well-being for one or more of the following PPIs: PERMA, *Three Good Things, Utilising your Signature Strengths in New Ways and/or Best Possible Selves* [42,43,44,45,46,47,48,49,50]. The *Best Possible Selves* intervention showed a statistically significant effect on reducing negative emotions post-intervention [45,46]. However, there was no statistically significant effect of *The Three Good Things* Intervention on mood [45,46,50] or mental well-being post-intervention [45,46]. These findings were inconsistent with other studies in the review which found a statistically significant effect of the *Three Good Things* intervention on well-being post-intervention [47,49].

Sustainability of the effect of the PPI at 3- and 6-months: (Appendix A presents the mean effect size of the PPI on well-being over time plus information about the intervention design and well-being measurements.): PERMA, *Utilising Your Signature Strengths in New Ways and Three Good Things* showed a statistically significant effect on well-being at 3-months [42,47,48,49]. At 6-months, three studies showed that *PERMA*, *Three Good Things and Utilising Your Signature Strengths in New Ways* showed a statistically significant effect on well-being [42,47,49].

The mean effect sizes demonstrated that the magnitude of the effect on well-being varied. Focusing on the sustainability of effect at 3-months, four studies showed that the PPIs PERMA, *Gratitude and Utilising Your Signature Strengths in New Ways* had a small mean effect size on well-being [42,47,48,49]. At 6-months, four studies reported a small effect on well-being for the PPIs *Gratitude Visit*, *PERMA*, *Utilising your Signature Strengths in New Ways, and Three Good Things* [42,47,48,49]. *Utilising Your Signature Strengths in New Ways* intervention [47] reported a medium effect size at the 6-month follow-up which showed the most promising impact on well-being.

Recruitment of Participants: Participants were recruited through various routes such as senior centres [44], word of mouth, posters, local newspapers, or community magazines and fliers [42,45,46,50]. Other studies employed recruitment strategies via online social media platforms such as Facebook [47] and online resources such as mailing lists, online discussion forums and media reports [43,45,46,48].

Quality of Methodology: Three studies were rated as high quality [42,43,48], four as medium quality [45,46,47,49] and two low in quality [44,50]. Recruitment of participants into the nine studies was also evaluated for suggestions on effective strategies to employ (see Appendix A for a summary of findings).

Diary: A PERMA-based diary was integrated into the intervention to be completed each day of the week over the 7-week timeframe. The diary served as an independent activity to complement the weekly sessions. Several studies were found in the positive psychology literature to support the adoption of a diary to encourage individuals to find the positive or joy in the mundane of life and to add a degree of reframing of how things are seen [52,53,54].

V1: In line with the findings from the positive psychology literature, the content of V1 included a broad range of other effective PPIs namely, *Three Good Things, Best Possible Selves*, *Utilising Your Signatures in New Ways*, and *Character Strengths*. The method of delivery was identified as being a one-to-one intervention with a facilitator with weekly 1 h consultations delivered for a period of 7 weeks. A PERMA-based diary was designed with space for daily entries to mark ‘yes/no’ if the PERMA element was experienced each day and space to explain how the element was experienced. The diary was to be completed every day for a period of 7 weeks.


*Step 2—Modelling Version 1/Gathering Feedback to Produce Version 2*


V1 was tested with fifteen individuals. The intervention was delivered face-to-face on a one-to-one basis (*n* = 7), group setting (*n* = 4) or group setting via WhatsApp video call (*n* = 4). Fourteen individuals completed the seven-week intervention, one dropped out at week four due to personal reasons. Although the purpose of this step was not to report the perspectives of the individuals who helped to model V1, engagement with the intervention was very positive which provided meaningful feedback for further consideration. For example, the timescale for the delivery of the intervention was reduced from 7 to 6 weeks in line with user feedback to align with the five dimensions of the PERMA model (week 1—introduction, weeks 2 to 6 -one aspect of the model covered per week) and to address time constraints relating to the setting, intervention deliverer and participants. Another key amendment was the reduction in the requirement for completing the diary from daily to 3 days (of their choice) per week and the modification of language to include clear definitions and examples for each PERMA element.

A summary of all changes made to V1 based on this feedback and discussion with the research team is presented in Table 2.


*Step 3—Revisiting the Evidence to Produce Version 3*


No new studies were identified when the literature search was repeated, however, minor aspects of the intervention were refined based on further review of V2 to produce V3 *Identification of Character Strengths* was introduced in the second facilitated session and the fifth session incorporated an activity where an individual’s signature strengths were revisited and compared to their core values.

The iterative approach adopted for the development of the intervention versions (V1, V2 and V3) with rationale for changes is presented in Table 2. The main changes focused on the design and content which included modifications to the activities, simplifying the language, and questions asked and completing one specific PPI per session. Additionally, the PERMA diary was simplified through the removal of the ‘yes/no’ question, an explanation of each PERMA element on a summary page and its completion reduced from every day to 3 days a week.

Version 3, the final version of the well-being intervention was entitled ‘*Prescribing Happiness’ (P-Hap)*, based on feedback from individuals and discussions with the research team. A full outline of the activities included in P-Hap is presented in Table 3. An example of Session 1 of the P-Hap is presented in Appendix A together with the PERMA-based diary. The TIDieR checklist [Appendix A] describes the transparency of reporting for the components of the intervention.

## 4. Discussion

The development of this positive psychology well-being intervention followed a robust iterative process as recommended by the MRC framework for complex interventions [36]. Three key steps of the Development, Phase 1, were completed, resulting in an evidence-based intervention titled *Prescribing -Happiness (P-Hap)* ready for the next stage of development with external stakeholders.

### 4.1. Step 1—Identification of the Evidence-Base and Modelling V1

The use of positive psychology constructs in the design of well-being interventions is a relatively new concept [57] which might explain why no studies were found using PPIs in a community pharmacy setting. However, the review of the literature, using a systematic approach provided insights from nine studies that investigated PPIs in community-based samples between January 2011 and March 2018 (extended in Step 3 to October 2019). These findings provide encouraging support for the delivery of PPIs in a community-based setting. It has been noted that the best practice for developing interventions is to utilise the best available evidence and appropriate theory in a systematic, rigorous, and robust way [58]. Being transparent on how complex interventions are developed is key to ensuring that evidence-based interventions can be properly replicated and adopted worldwide [35]. All nine studies reported a statistically significant effect of the PPIs on well-being from pre-intervention to post-intervention. These studies utilized either PERMA, *Three Good Things, Utilising Your Signature Strengths in New Ways and/or Best Possible Selves* either alone or in combination [42,43,44,45,46,47,48,49,50]. Importantly, PPIs which were developed from positive psychology theories and/or models were effective in improving well-being and eliciting positive emotions and/or decreasing negative emotions [42,43,44,45,46,47,48,49,50]. The PERMA model was identified as a suitable underpinning theoretical basis of the intervention design since it formed the basis of PPIs in numerous community settings [42,43,44,45,46,47,48,49,50] and was shown to be effective in enhancing well-being in the short and long term [42]. In addition, PERMA and PPIs has been extensively used in a range of other settings such as healthcare, education, sport and in the workplace [59,60,61,62].

Some PPIs were delivered over a 1-week timeframe [42,43,45,46,47,48,49], however, this short timeframe limited the variety of components that could be included, and the length of time for individuals to learn, apply, and reap the benefits from engaging with a range of PPIs. Knowing the sustainable effect of a public health intervention is of great importance [63]. If the effect wanes over time, this can lead to a waste of time and resources which brings into question the benefit of engaging with the intervention. The PPIs— ‘PERMA’, *‘Three Good Things’ and ‘Utilising Your Signature Strengths in New Ways’* demonstrated statistically significant effects on well-being at 3-months and 6-months [42,47,48,49]. The mean effect size showed a small effect (Cohen’s d = 0.2) on well-being at 3-months for PERMA, Utilising Y*our Signature Strengths in New Ways, Three Good Things and Gratitude Visit* [42,47,48,49] and a moderate (Cohen’s d = 0.5) effect at 6 months for ‘*Utilising Your Signature Strengths in New Ways’* [47]. These findings highlight the need for multiple interventions or ‘multiple-component PPIs’ where possible. The benefits of including a variety of evidence-based activities targeting two or more hedonic and eudaiomonic well-being components has also been emphasized [57]. V1 of the intervention was therefore designed as a 7-week intervention to ensure each element of the PERMA model was covered plus other additional core aspect of positive psychology, namely, *’Character Strengths’*, *‘Signature Strengths in New Ways’*, *‘Three Good Things’*, *Gratitude’ and ‘Best Possible Selves’*.

Recruitment, choice of outcomes and improvements in retention are three other priorities when considering the appropriate methodology in research [64,65,66,67,68]. PPIs identified from this review were delivered effectively using a diverse range of methods such as online, in a group or on an individual basis with a facilitator. All studies had varied recruitment strategies inclusive of social media, local newspapers, and a community centre. Eight out of the nine studies delivered the PPIs online. Attrition rates varied from 0% to 60% across studies. Incentives were used across all studies, for example financial and personalized feedback on well-being. Interestingly, the two studies which reported the lowest attrition rates offered no incentives [45,46]. Seven studies were RCTs [42,43,45,46,47,48,49]. Recruitment to RCTs has numerous challenges which can impact upon the cost and delivery of the trial [69]. Prioritising how participants were recruited and employing a number of strategies to maximise participant retention were key factors to consider [66]. Inequalities in digital health access must be kept in mind as some groups in society are unable to reap the benefits of digital technologies [70]. Therefore, the design of an appropriate recruitment strategy needed to consider that a proportion of the population may not have access to online platforms.

One approach found to be effective for recruiting in primary care settings has been to involve the healthcare practitioner in the process and this has also been the case for PPIs studies [71]. However, none of the studies included in the current review employed any type of HCP as part of their recruitment process. Word of mouth from HCPs (or friends/family) was also reported to be one of the most effective strategies for raising awareness of community pharmacy services [72]. HCPs such as community pharmacists play a key role in providing information and public health services to their local community members. To date, the most effective recruitment strategy for community pharmacy-based public health interventions or mental health trials remains unclear [67,73]. Therefore, a recruitment strategy was needed that ensured equal, open, and accessible approaches while taking the demographic characteristics of the local population into account.

It was not possible to make any conclusions as to whether a PPI administrated to a group is more effective than a PPI delivered on an individual basis. Only one study was conducted as a face-to-face intervention (90 min group sessions once a week) [50], all others were administered online and completed without any facilitator support. An individual intervention has been recommended for community-based samples which are recruited via a healthcare practitioner and delivered on an individual basis [71]. Group interventions have been recognised as potential cost-effective approaches as they treat more than one person at a time [68]. However, from a well-being perspective, certain populations may not be suited to group interventions and there may be trust issues which arise within the group dynamics. Additionally, from a pragmatic perspective, many healthcare settings may not be able to facilitate a group intervention due to space. Community pharmacy teams have expanded their roles greatly over the last 20 years with the requirement for consultations rooms away from the public area of the pharmacy, enabling the delivery of many new services [74]. The infrastructure and experience of a community pharmacy in delivering one-to-one services was seen as an enabler for a one-to-one well-being intervention in this setting.

Understanding the context, namely, the setting and environment, is vital to the development of an intervention and the process of implementation [35,75,76,77,78,79]. The process of undertaking observational research enabled coproduction during the research process [79]. Interventions which are developed through coproduction between practitioners, researchers, service providers and members of the public can maximize their effectiveness [75,76,78]. A lack of understanding around the intervention context has been identified as contributing to the critical gap between research and practice [79]. The development of *P-Hap* benefited from the careful consideration of how the community pharmacy context shapes the interventions. Based on the findings of this review, and the observational notes of the researcher, it was agreed that an individual consultation would be the best choice for delivery of a PPI in a community pharmacy setting, either face-to-face or online.

Diary writing or structured expressive writing has numerous benefits for psychological and physiological health, health behaviours, and perceived physical health [23,52,54]. Structured writing differs from expressive writing as it asks participants to write about a specific event or experience for a set time on a daily, weekly, or monthly basis. The purpose of the diary was to guide individuals towards focusing on what they already do to support their well-being rather than needing to necessarily change what they do. It therefore encourages people to see their lives through a new lens, advocating savouring, slowing down and attracting attention to simple PERMA-based moments and activities in one’s day-to-day life.

Diary writing or structured expressive writing has numerous benefits for psychological and physiological health, health behaviours, and perceived physical health [23,53,55]. Structured writing differs from expressive writing as it asks participants to write about a specific event or experience for a set time on a daily, weekly, or monthly basis. The purpose of the diary was to guide individuals towards focusing on what they already do to support their well-being rather than needing to necessarily change what they do [64]. It therefore encourages people to see their lives through a new lens, advocating savouring, slowing down and attracting attention to simple PERMA-based moments and activities in one’s day-to-day life.

### 4.2. Steps 2 and 3—Modelling of the Intervention and Revisiting the Literature

A range of individuals known to the research team shared their perspectives on the design and content of the intervention including the structure and delivery to support the evolution from V1 to V2. These represented experts in positive psychology and intervention design, pharmacy staff, and members of the public. They provided rich feedback which contributed towards the modelling process and iterative development of V1 to V2. At this stage, the completion of the diary was reduced from daily to three times a week to accompany the weekly one-to-one consultation over a 6-week timescale for the delivery of the PPI.

No further studies were identified from revisiting the literature. V3 was therefore produced based on some minor modifications to V2 which were mostly related to clarification of language. This aligns with UK government policy which highlights the importance of providing good quality health information to encourage better patient participation in self-management of long-term conditions and promotion of well-being [80]. Printed educational material is simple, easy to implement and relatively inexpensive for the dissemination of information but it is important to write with clarity, use large fonts, avoid jargon, and explain any terminology [81,82,83]. Based on the modelling feedback and following this guidance, the PERMA-based diary and content of sessions were modified to ensure clear language and simple terminology.

It was agreed that the resultant V3 of the intervention would be titled, *‘Prescribing Happiness (P-Hap)*’. Although the term ‘happiness’ does not address the eudaiomonic aspect of well-being, it was felt that this term would be the most appropriate for use with the community population of pharmacy customers.

### 4.3. Strengths and Limitations

A strength of this research was the successful adoption of a rigorous and systematic approach to the effective intervention design for implementation in a real-life setting [84]. The iterative development process undertaken, following the MRC framework, and taking into account observations of the context setting are considered to be key strengths of this study.

However, some limitations need to be considered. When conducting the literature review, there did not seem to be an accepted definition of the term ‘community’ which was used differently across disciplines. Only two studies clearly mentioned that they specifically utilised community-based samples [44,50] making study selection challenging. Most of the participants across all nine studies were female; however, this is a common theme across well-being studies which is widely acknowledged [42,44,46,49]. Nonetheless, the findings yielded useful data for the design of a PPI for use in a community pharmacy setting, particularly considering that women are more likely to use pharmacies than men [85]. Future recruitment strategies need to consider how male participants’ engagement in PPIs can be improved, possibly through the use of existing interpersonal relationships or social networks with females undertaking the programme.

Public and patient involvement is widely recognised as being beneficial during the development of health interventions [86]. Intervention design can sometimes fail to recognise the needs of the target user, thus engaging the public and patients can improve the relevance and quality of research [86]. Whilst this study involved feedback from lay members of the public, only pharmacy staff from the same geographic location were involved in the modelling process. The process of designing the P-Hap may have been strengthened by engaging service users and members of the public from the local population during the modelling process. However, it is important to note that the ethnographic and observational activities undertaken before commencing this research and the involvement of the pharmacy staff helped to make sure that the intervention was culturally sensitive to the needs of the local population so that the *P-Hap* was fit for purpose for the community pharmacy service users. Most pharmacy staff perspectives were gathered from those who worked within the research setting and this may limit the suitability of the intervention for use in other community pharmacy contexts in the future, especially since this study setting may not be reflective of the business model for all community pharmacies in the UK. Whilst community pharmacies in the UK do not keep documentation on information such as socio-economic, health literacy or cultural background, they have a good understanding of the social and cultural characteristics of their service users. Therefore, members of the pharmacy team are often representative of the local population and should be involved in the adaptation of the *P-Hap* to ensure the use of relevant PERMA examples and appropriate language.

### 4.4. Recommendations for Policy, Practice and Research

Decision makers at policy level need sufficient information on the content and delivery of an intervention to allow it to be replicated [87]. Primary care settings such as GP surgeries and community pharmacies deliver a range of public health interventions to support community health and well-being, yet no PPIs were delivered from these settings were in the published literature. These settings should be utilised to recruit community members and deliver PPIs due their familiarity and accessibility within communities. More research is needed to gather the insights of key stakeholders working in these settings (for example, GPs, Practice Nurses, Community Wellbeing Coordinators, Public Health Specialists) to establish the feasibility of implementing PPIs through existing healthcare services.

A key aspect of the modelling stage was the use of multi-components in the design of the P-Hap intervention since these PPIs were shown to be more effective than those that adopted only one PPI activity [19,60,88]. Since PERMA is the only multi-component model in positive psychology, it is recommended that this is adopted when designing a PPI for use in a community setting. Complex interventions are built upon numerous components including behaviours, time, dose, duration, methods of organising and delivery (i.e., practitioner, setting, location), all of which may act independently or interdependently [76]. Completion of the TIDieR checklist allowed full transparency of the P-Hap intervention to mitigate against any uncertainty and to ensure that the intervention can be implemented in practice [89].

Another output of the review was the importance of employing multiple recruitment and retention strategies in community-based PPI research. Therefore, a multi-method approach to recruitment and retention is clearly needed when conducting research in community-based PPIs.

PPIs of longer duration (4 to 8 weeks) are to be endorsed when developing a well-being intervention. It is recommended that the sustainable effects on well-being are measured up to 3- and 6-months post-intervention. Heintzelman and Kushlev [90] state that positive psychology needs to move away from the use of self-report measures and mono-research designs to include both qualitative and quantitative research methodologies. This review found that only one study included a qualitative methodology [44]. Therefore, more qualitative research is needed to enable investigation into ‘real life’ examples of the effectiveness of PPIs which draw on participants’ experiences in-depth.

The next stage of the research is to gather the views of a variety of external stakeholders from a practice and policy perspective, on the feasibility of implementing the P-Hap intervention in a community pharmacy setting. This will enhance the future transferability of the intervention into community pharmacies and the potential to be implemented into similar context settings in the future.

## 5. Conclusions

Current health and public health policies in the UK recommend moving towards the integration of well-being interventions to augment existing NHS services. The *Prescribing Happiness (P-Hap)* intervention evolved from an iterative, and robust, systematic process involving a mixture of theory, modelling, and practical-based approaches. Nine studies were identified which informed the modelling stages of P-Hap development. Based on a review of the evidence base for the effectiveness of community-based PPIs, it can be concluded that the design of PPIs should move away from the standardized approach of using one PPI, towards interventions which incorporate multiple PPIs. These should be delivered over 4 to 8 weeks and the sustainable effects on well-being should be measured at 3 and/or 6 months. Therefore, the PERMA model of positive psychology was used to underpin this community pharmacy-based PPI with the inclusion of five other core aspects of positive psychology, namely *’Character Strengths’, ‘Signature Strengths in New Ways’, ‘Three Good Things’, Gratitude’ and ‘Best Possible Selves’*. The next stage of this research is to gather the perspectives of key external stakeholders, such as policy makers, community pharmacy staff, GPs, and well-being facilitators, on the feasibility of implementing *P-Hap* in a community pharmacy setting to improve the well-being of the local population.

## Figures and Tables

**Table 1 pharmacy-11-00014-t001:** Outline of the PERMA well-being model used to underpin the positive psychology intervention.

PERMA Well-Being Element	Description
Positive emotions	Positive emotions are associated with feelings of contentment, hope, pleasure, comfort, gratitude, and joy.
Engagement	Engagement is a deep psychological connection to an activity which stimulates a state of concentration on an intrinsically motivating task. Engagement has also been conceptualized as the ‘absorption’ and feeling within a state of ‘flow’ when completely immersed in an activity.
Relationships	Relationships are central to feeling connected, integrated as part of a community, and being cared for by loved ones.
Meaning	Meaning refers to the sense of feeling connected to something larger than oneself.
Accomplishment	Accomplishment is a feeling one gets after perseverance towards a goal or future outcome.

**Table 2 pharmacy-11-00014-t002:** Summary of changes to content and design of the well-being intervention for Versions 1, 2 and 3.

Change to:	V1 Design/Content (Step 1)	Rationale for Changes to V1 (Step 2)	V2 Design/Content	Rationale for Changes to V2 (Step 3)	V3 Design/Content
**Diary**	**Design**: Open questions for how each PERMA was experienced each day for one week.	**Content**: Tick boxes ‘Yes/No’ removed. Open Question- too long and confusing. Include a summary page with PERMA examples.	**Content**: Enter PERMA element description for 3 days a week instead of daily.	**Content and Design**: No Change	**Content and Design**: No Change
**1st Session**	**Content**: Title ‘IntroductionSession’	**Content**: Need friendly approach with use of layman terms	**Content**: Title changed to ‘Taster Session’	**Design**: New 1st session of the intervention.	**Content and Design**: No Change
**2nd** **Session** **Activity 2**	**Content**: Your story of overcoming a challenge. *Identify your Character Strengths*	**Content**: The ‘challenge’ aspect was emotionally difficult	**Content**: *You at Your Best* & *Identification of Character Strengths* from *You at Your Best* Story	**Design**: Not enough time to complete exercise. **Content**: Complications over choosing *You at Your Best* example	**Content**: *Identification of* *Character Strengths* in General Life
**3rd** **Session—Activities 1 and 3**	**Content**: Engagement and Character Strengths. Area of life forengagement	**Content**: Removal of Character Strengths section- no specific connection. Detailed questions were confusing	**Content**: Area of life for engagement- why, what focused questions	**Content**: Detailed questions were confusing. Need to simplify questions	**Content**: Activity 3: Select one engagement activity only
**4th Session** **Activities 1 and 2**	**Content**: What relationships are you grateful for and why? Linking core values and engagement to relationships	**Content**: Removal of core values and engagement activity. A stronger evidence-based approach needed	**Content**: Simplified Gratitudeintervention and Why? Random Acts of Kindness	**Content and Design**: No change	**Content and Design**: No change
**5th Session**	**Content**: Identification of core values and reason why?	**Content and Design**: No change	**Content and Design**: No change	**Content**: Need to add other interventions which link to values.	**Content**: Identification of values and matching values to Signature Strengths in session 2
**6th Session Activity 1**	**Content**: Goal Setting (What, why and how questions?)	**Content**: Questions were not specific and caused confusion	**Content**: Goal Setting (What is the Goal, What Steps going taken and three reasons why?)	**Design**: Took too long to complete goalsetting activity	**Content**: *Best Possible Selves*

**Table 3 pharmacy-11-00014-t003:** Description of *Prescribing Happiness (P-Hap)* intervention activities with supporting references.

Week	PERMA	Workshop Title	Intervention(s)	Reference	Intervention Description
**1**	Character Strengths	Character Strengths	Introduction and *Identification of* *Signature Strengths*	Adapted from [55]	Identification of qualities and strengths. Participants are asked to write down their five Signature Strengths from the 24 Character Strengths.
**2**	Positive Emotions	Positive Emotions and Gratitude	Happiness Toolkit, *Three Good Things*	Adapted from [19,20,21]	Identification of six things which make happy. Participants asked to write Three Things they are grateful for and why.
**3**	Engagement	Engagement and Flow	Identification of flow activities	-	Identification of activities which create a state of flow. Participants asked to complete one new flow activity before the next session.
**4**	Relationships	Relationships	Three Relationships Grateful For and Random Acts of Kindness	Adapted from [52,56]	Identification of people they are grateful for and why. Participants are asked to perform five acts of kindness for family, colleagues, friends, community, and selves.
**5**	Meaning	Meaning and Core Values	Identification of Values and LinkingSignature Strengths and Value	Adapted from [55]	Identification of five core values. Participants asked to match their five core values to their signature strengths (Week 1) & explore why they match.
**6**	Accomplishment	Accomplishment	*Best Possible Selves*	[23]	Participants asked to write their *Best* *Possible Selves* in five years’ time.

## Data Availability

Not applicable.

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
