# Peer review of "Development of a Positive Psychology Well-Being Intervention in a Community Pharmacy Setting"

_pharmacy, 2023, doi:10.3390/pharmacy11010014_

Round 1

Reviewer 1 Report

This paper described the 3-step development of a community pharmacy-based psychological well-being intervention in South Wales. The authors did a great job presenting the background and needs. The descriptions of the 3-step process for intervention development were well done as well. I only have a few minor suggestions for the authors to consider in order to improve the quality of their study.

1. Has the PERMA model being validated in the welsh population? If yes, the authors should cite previous studies.
2. The authors might consider strategies to improve male participants engagement in the intervention in the future. Possible strategies could be through interpersonal relationship, i.e., encouraging female participants to bring the males in their family/social network to join them.

3. Line 531, the authors should specify what stakeholders could be involved in the next steps. 

Author Response

Reviewer 1 , Comment 1 - Has the PERMA model being validated in the welsh population? If yes, the authors should cite previous studies.

Authors' response - this is indeed a good question. Whilst the PERMA model / profile has been validated in other countries such as the USA, Australia, this has not been undertaken for the population of Wales. There are emerging studies of its use in the educational and sporting sectors, and also translation into the Welsh language this body of work is currently unpublished.

Reviewer 1, Comment 2 - The authors might consider strategies to improve male participants engagement in the intervention in the future. Possible strategies could be through interpersonal relationship, i.e., encouraging female participants to bring the males in their family/social network to join them.

Authors' response - we thank the reviewer for this useful consideration which has now been added to the Discussion (Line 523-524) as a suggestion for future research.

Reviewer 2 - Comment 3 - Line 531, the authors should specify what stakeholders could be involved in the next steps. 

Authors' response - the line has now changed to 550-552 due to earlier additions. We have now added some examples of stakeholders to this sentence.

Reviewer 2 Report

Your proposal is very interesting, but I think your presentation could be improved, as in its current formulation it is a mixture of two different procedures that are part of the same project. I suggest that, starting from the same introduction, you divide the work (this can be done from the methodology) into two different studies: the literature review and the pilot study on the implementation of the intervention. In my opinion, this would be more comprehensible for the potential reader.

I have no suggestions regarding the systematic review, except that I could be more explicit about the search terms and their combination. Regarding the pilot study on the implementation of the intervention, I think it would be good to know a bit more about the potential users. With the data provided on the target community, it is impossible to determine whether the intervention is adapted to the community's capacity for change and its human diversity. For example, has the cultural diversity of the potential recipients of the intervention been considered, and what measures have been taken in this regard? It is essential that these interventions be culturally sensitive. That is, they should consider and incorporate into their content and materials the ethnic and cultural characteristics, values, behaviours, and beliefs of the target population, as well as relevant social, environmental and historical factors. I believe it is necessary to clarify what measures have been or will be taken in this respect. Finally, I hope these suggestions will be useful to improve the exposure of your work, which, I believe, is innovative and very interesting for improving the health of the general population and enhancing the development of positive psychology.

Author Response

Reviewer 2, Comment 1 -Your proposal is very interesting, but I think your presentation could be improved, as in its current formulation it is a mixture of two different procedures that are part of the same project. I suggest that, starting from the same introduction, you divide the work (this can be done from the methodology) into two different studies: the literature review and the pilot study on the implementation of the intervention. In my opinion, this would be more comprehensible for the potential reader.

Author’s Response - Thank you for your positive feedback on our paper. To clarify, no pilot study was undertaken, and the intervention was not implemented in practice during this stage of the research. This study was focused on the initial modelling and development phase of creating a positive psychology intervention suitable for delivery in a community pharmacy setting. The development process was followed in accordance with the MRC Framework for complex interventions (Skivington, et al, 2021) which does not recommend piloting or implementing an intervention in practice during the stages of development. We have therefore followed the MRC framework guidelines for Phase 1 ‘Development’ both when conducting and reporting these stages of the research (i.e., gathering the evidence base and modelling process) for transparency and accuracy. We have added a note on this to the end of methods section, in order to help the potential reader who may be less familiar with this recommended approach. We hope you agree that this is informative for the reader.

Reviewer 2, Comment 2 –

a) I have no suggestions regarding the systematic review, except that I could be more explicit about the search terms and their combination.

Author’s Response –

a) We agree with your feedback on this and we thank you for pointing this out. The search terms and their combinations have now been added as a Table to  Supplementary material 1 to provide clarity.

b) Regarding the pilot study on the implementation of the intervention, I think it would be good to know a bit more about the potential users. With the data provided on the target community, it is impossible to determine whether the intervention is adapted to the community's capacity for change and its human diversity. For example, has the cultural diversity of the potential recipients of the intervention been considered, and what measures have been taken in this regard? It is essential that these interventions be culturally sensitive. That is, they should consider and incorporate into their content and materials the ethnic and cultural characteristics, values, behaviours, and beliefs of the target population, as well as relevant social, environmental, and historical factors. I believe it is necessary to clarify what measures have been or will be taken in this respect.

 Author’s Response –

b) With regards to the request for further information about the potential users, this was something that was considered in depth at each stage of the research in discussion with the Company Partner i.e. owners of the community pharmacy (AW, MR) and wider research team. Before commencing the study, the researcher (JW) spent extensive time in the pharmacy drawing on ethnographic and observational approaches to ensure a good understanding of the study population before the intervention was designed to ensure that it was fit for purpose for the community pharmacy service users.

Whilst no information such as socio-economic, health literacy, cultural background is kept on record at the pharmacy, they have a good understanding of the demography and wider characteristics of their service users which was considered in each step of development.  We took every possible measure to ensure that members of the pharmacy staff who took part in Phase 1 were representative of the local population and provided their feedback on aspects of literacy, simplicity of language and the use of appropriate PERMA examples. Their perspectives were incorporated into the P-Hap materials to ensure the intervention was culturally sensitive, and the content was of relevance to the local community. This is highlighted in Table 2 where the PERMA-based diary and worksheets were redesigned into a simpler format to ensure it was suitable for members of the public.

We have now added a section to the Discussion to make sure that the importance of the activities described above are made clear to the reader and also how these issues should be considered in any further research or adaptation of the P-Hap. We would like to thank Reviewer 2 for reminding us of the importance of highlighting these crucial points.

c) Finally, I hope these suggestions will be useful to improve the exposure of your work, which, I believe, is innovative and very interesting for improving the health of the general population and enhancing the development of positive psychology.

 Author’s Response –

c) Thank you again for your encouraging and positive words which are greatly appreciated. We would like to take this opportunity to thank Reviewer 2 once again for the constructive and helpful suggestions which will strengthen the robustness and readability of the paper.

Reviewer 3 Report

The paper is very interesting and well-written.

Referring to the studies and EU actions... the role of the "healthcare facilities" on the territory can be strategic.

The method is clear.. and the entire work has a good sound.

From my point of view.. the paper can be published.

Author Response

Reviewer 3  - We would like to take this opportunity to thank you for reviewing our paper and for providing such positive comments.
